# Prognostic Factors Analysis for Intracranial Cavernous Malformations Treated with Linear Accelerator Stereotactic Radiosurgery

**DOI:** 10.3390/life12091363

**Published:** 2022-08-31

**Authors:** Meng-Wu Chung, Chi-Cheng Chuang, Chun-Chieh Wang, Hsien-Chih Chen, Peng-Wei Hsu

**Affiliations:** 1Department of Neurosurgery, Chang Gung Memorial Hospital at Linkou, Chang Gung University, Taoyuan 333, Taiwan; 2Department of Radiation Oncology, Chang Gung Memorial Hospital at Linkou, Chang Gung University, Taoyuan 333, Taiwan; 3Department of Neurosurgery, Chang Gung Memorial Hospital at Keelung, Chang Gung University, Taoyuan 333, Taiwan

**Keywords:** adverse radiation effect, cavernous malformation, edema, linear accelerator, radiosurgery, rebleeding

## Abstract

Stereotactic radiosurgery (SRS) is generally considered a substitute for cranial cavernous malformations (CCMs). However, prognostic factors for post-radiosurgery CCM rebleeding and adverse radiation effects have not been well evaluated, and the effect of timing and optimal treatment remains controversial. Therefore, this study evaluated prognostic factors for post-radiosurgical rebleeding and focal edematous changes in 30 patients who developed symptomatic intracranial hemorrhage due to solitary non-brainstem CCM and received linear accelerator (LINAC) SRS in a single medical center from October 2002 to June 2018. An overall post-radiosurgical annual hemorrhage rate with 4.5% was determined in this study. In addition, a higher marginal dose of >1600 centigray and earlier LINAC SRS intervention were correlated with a significantly lower post-radiosurgical annual hemorrhage rate. A lesion size larger than 3 cm^3^ and a coexisting developmental venous anomaly were significant risk factors for post-radiosurgical focal brain edema but mostly resulted in no symptoms and were temporary. This study demonstrated the efficacy of LINAC SRS in preventing CCM rebleeding and suggests that earlier radiosurgery treatment with a higher dose for non-brainstem symptomatic CCMs be considered.

## 1. Introduction

Cerebral cavernous malformations (CCMs) are clusters of dilated sinusoidal channels lined with endothelial cells that do not intervene in the brain parenchyma or large feeding or draining vessels [1]. They are the second most common type of vascular lesions in the central nervous system, comprising 10% to 15% of all vascular malformations [2,3]. Despite their low-flow characteristic, CCMs can cause symptomatic intracranial hemorrhage, seizure, and focal neurological deficits. Symptomatic hemorrhage is the most severe complication of CCMs, and the prevention of rebleeding and hemorrhage-associated complications is the primary objective of treatment.

Although surgical resection remains the definitive treatment, stereotactic radiosurgery (SRS) has become increasingly crucial for treating CCMs located in deep brain areas, where surgery may result in a high risk of complications [4,5]. However, although the consensus indicates a clustering tendency of CCM hemorrhage [6,7,8], opponents of SRS suspect that the decreased annual hemorrhage rate (AHR) after SRS may only reflect the natural course of CCMs [9,10]. In addition, the optimal marginal dose of SRS for treating CCMs remains uncertain because of the risk of subsequent transient or persistent adverse radiation effects (AREs). Thus, patient selection, risk factor identification, and dose selection are crucial when making decisions regarding microsurgery, SRS, or conservative treatment.

In this single-center retrospective study, we present our experience with 30 patients diagnosed as having a solitary, non-brainstem, repeated-bleeding CCM treated with linear accelerator (LINAC) SRS to evaluate factors affecting treatment outcomes, post-radiosurgical rebleeding, and SRS-induced perifocal brain edema.

## 2. Materials and Methods

### 2.1. Setting and Design

This retrospective cohort study evaluated prognostic factors for post-radiosurgical rebleeding and focal edema in a single medical institution. Treatment planning, medical history, clinical presentation, and image studies were retrieved from medical records. The institutional review board approved the study and agreed to waive the requirement for informed consent (IRB No. 201700082B0 obtained on 8 February 2017).

### 2.2. Patient Population

Patients with sporadic CCMs who experienced repeated bleeding events were included if they had received LINAC SRS in our hospital between October 2002 and June 2018. Patients were excluded if (1) they had more than one CCM because the characteristics of familial-type CCMs differ from those of sporadic CCMs [11], (2) underwent prior interventions for CCM, and (3) had underlying cognitive impairment, which would result in low reliability of the outcome assessment. Follow-up data were collected through a chart review. Before treatment, each patient went through an interdisciplinary board discussion. The committee was made up of neurosurgeons, neuro-radiologists, and radiation oncologists.

### 2.3. Radiosurgical Techniques

One-millimeter-thick slices obtained through enhanced computed tomography (CT) and T1-weighted with contrast, T2-weighted, and/or Fluid-attenuated inversion recovery (Flair) magnetic resonance imaging (MRI) (1.5 tesla, Optima^TM^ MR450w GEM, GE healthcare, Chicago, IL, USA) from the vertex of the skull to the foramen magnum, scanned without a gap, were transferred through the network to the software Brainscan, version 5.31 (Brainlab, Munich, Germany) for treatment planning. The gross target volume (GTV) was calculated using the iPlan Radiotherapy Planning Software (iPlan RT Image 3.0.1, Brainlab, Munich, Germany). SRS was conducted using the Novalis treatment system (Brainlab, Heimstetten, Germany). The dose applied to the periphery of the tumor ranged from 1500 to 2250 centigray (cGy), and a 100% isodose line was employed to encompass the tumor margin (Figure 1).

### 2.4. Clinical and Image Follow-Up

Following LINAC SRS, follow-up MRI was performed in the 3rd and 12th months and then annually after treatment. CCMs were classified as superficial if the gray and white matter junctions were involved and deep if the junctions were not involved. Volumetric analysis was performed using a T2-weighted MRI study on the iPlan radiotherapy planning software. The Zabramski classification was used to classify CCMs on the basis of the MRI characteristics [12]. Type I CCM was defined as subacute hemorrhage with hyperintensity on a T1-weighted image (T1WI) and either hypo- or hyperintensity on a T2-weighted image (T2WI). Type II CCM was defined as mixed hemorrhage (classic “popcorn” appearance), with a mixed signal intensity on both the T1WI and T2WI. Type III CCM was defined as chronic hemorrhage, with hypo- to iso-intensity on a T1WI and hypo-intensity on a T2WI. Type IV CCM was defined as multiple punctate microhemorrhages, which were difficult to identify on either a T1WI or T2WI. Eloquent locations were defined as the motor or sensory cortices, visual pathway or center, speech center, internal capsule, basal ganglia, and hypothalamus or thalamus.

### 2.5. Outcome Assessment

Rebleeding events were defined as new hemorrhages developed after SRS treatment and observed on CT or MRI, and were classified into symptomatic and non-symptomatic events. A symptomatic rebleeding event was documented if clinical symptoms were noted, including headache, seizure, consciousness disturbance, and focal neurological deficits compatible with image findings. Newly found hematomas in imaging studies without clinical symptoms and signs were defined as asymptomatic rebleeding. The AHR was calculated as total symptomatic hemorrhagic episodes divided by the total years of follow-up.

To evaluate the response in terms of volume control, the volume variation ratio (VVR) was defined as the lesion volume at the follow-up divided by the pre-radiosurgical lesion volume. The response was categorized as complete response (CR), partial response (PR), stable disease (SD), or progressive disease (PD). CR was defined as the complete disappearance of the CCM. PR was defined as at least a 30% decrease in the VVR of target lesions. SD was defined as a change between a 30% decrease and a 20% increase in the VVR. PD was defined as an increase of 20% or more in the VVR. CCMs with acute bleeding events were excluded from volumetric analysis because the measurement of the precise volume was difficult.

Post-radiosurgical perifocal brain edema events were defined as new-onset high signal intensities on T2-weighted MRI. A perifocal brain edema was defined to be persistent if the high signal intensity still remained on the last MRI follow-up. Similar to rebleeding events, perifocal brain edema events were classified into symptomatic and asymptomatic episodes on the basis of patients’ clinical presentations. If simultaneous rebleeding and perifocal edema were observed, the edematous change was considered to be induced by hemorrhage instead of radiation. However, if the edematous change persisted for more than 1 year after the rebleeding event, it was still considered to be induced by radiation.

### 2.6. Statistical Analysis

Prognostic factors were investigated for post-radiosurgical rebleeding and perifocal edema events in this study, including all symptomatic and asymptomatic events. Several variables including age, sex, marginal dose, deep lesion location, lesion volume, the time between the first bleeding episode and radiosurgery, and patients’ underlying comorbidities were investigated. A cutoff of marginal dose was assessed from the lowest (1500 cGy) with increasing intervals of 100 cGy. A cutoff of lesion volume was considered 1 and 3 cm^3^ for post-radiosurgical perifocal edema.

Univariate analysis was performed through Poisson regression to evaluate prognostic factors for post-radiosurgery rebleeding. To determine prognostic factors for post-radiosurgical focal brain edema, the unadjusted hazard ratio was analyzed using Cox proportional hazards regression. The *p* value was determined using the Kaplan–Meier analysis and the log-rank test for categorical variables and the Cox proportional hazards regression for continuous variables.

Variables with a *p* value of <0.2 in the univariate analysis were introduced into a multivariable regression model with a backward elimination method. Poisson regression was performed for post-radiosurgery rebleeding, and Cox proportional hazard regression was conducted for post-radiosurgery perifocal edema. A *p* value of <0.05 was considered statistically significant. Data analyses were conducted using SPSS version 22 (IBM Corp., Armonk, NY, USA).

## 3. Results

### 3.1. Patient Selection and Characteristics

A total of 59 patients who were diagnosed as having CCMs and underwent LINAC SRS at our hospital between October 2002 and June 2018 were included. Of these, 29 patients were excluded; 10 had brainstem CCMs and were treated with a protocol with a different dose-fraction formula, 3 were asymptomatic but still treated with LINAC SRS to suit the patients’ preferences, 15 had multiple CCMs that possibly were of the familial type and might have different characteristics from those of sporadic CCM [13], and 1 had underlying Sturge–Weber syndrome and cognitive impairment. Most of the remaining 30 tumors had Zabramski class II lesions in non-eloquent locations (83.3%). The median volume was 0.97 cm^3^ (range: 0.23–2.95 cm^3^). Motor deficit (36.7%) was the most common symptom at diagnosis, followed by headache (26.7%) and seizure (26.7%). LINAC SRS was applied to the cavernomas after hematoma absorbed completely. The process usually took several months. Most patients received the SRS treatment within 2 years after the bleeding event. The median interval between the first symptomatic hemorrhage event and radiosurgery treatment was 11 (range: 2–72) months, with a median marginal dose of 2000 (range: 1500–2250) cGy. The median clinical follow-up duration was 61 (range: 5–215) months. Table 1 lists the characteristics of the participants. 

### 3.2. Post-Radiosurgical Rebleeding Analysis

A total of nine rebleeding events were noted during the follow-up period, namely eight symptomatic events and one asymptomatic event (Figure 2). Among the eight symptomatic events, six occurred within the first 2 post-radiosurgical years, one occurred in the 3rd post-radiosurgical year, and the last one occurred in the 9th post-radiosurgical year. The only asymptomatic rebleeding event occurred in the 3rd post-radiosurgical year. The post-radiosurgical AHR was 4.5% overall, 10.9% within the first 2 years, 1.4% after 2 years, and 1.1% after 5 years (Figure 3). Two patients underwent the subsequent surgical removal of CCMs because of symptomatic rebleeding. 

Prognostic factor analyses were performed. For all rebleeding events (both symptomatic and asymptomatic), the univariate analysis revealed old age and a higher marginal dose to be potentially protective and a larger volume and a longer interval between the first bleeding event and radiosurgery to be potentially hazardous, each with a *p* value of <0.2. In terms of the cutoff value, a marginal dose of more than 1600 cGy was considered potentially protective. After confounding with multivariate Poisson regression models, a marginal dose of >1600 cGy (*p* = 0.033) and a shorter interval between the first bleeding event and LINAC SRS (*p* = 0.020) were observed to exert significant protective effects for post-radiosurgical rebleeding prevention. Similar results were observed for symptomatic rebleeding events. The results are summarized in Figure 4 and Figure 5.

### 3.3. Response in Volume Control

A total of 26 participants were included in the volumetric analysis. Four participants were excluded. Of these four participants, three did not have pre-radiosurgical MRI data for lesion volume comparison because of loss of data from the hospital computer memory. One participant was lost to follow-up after a repeated bleeding event 3 months after receiving LINAC SRS treatment and did not undergo further MRI study. 

Within the 1st post-radiosurgical year, one participant did not undergo MRI follow-up. None of the 26 CCMs reached CR, 5 (19.2%) reached PR, 17 (65.4%) reached SD, and 3 (11.5%) showed PD. Within the 2nd post-radiosurgical year, none of the CCMs reached CR, 10 (38.5%) reached PR, 13 (50%) reached SD, and 3 (11.5%) showed PD. Within the 5th post-radiosurgical year, none of the CCMs reached CR, 10 (38.5%) reached PR, 13 (50%) reached SD, and 3 (11.5%) showed PD. Within the 10th post-radiosurgical year, 2 (7.7%) of the CCMs reached CR, 12 (46.2%) reached PR, 11 (38.5%) reached SD, and 1 (3.8%) showed PD. The two CCMs reached CR at the 6th and 7th post-radiosurgical years, respectively (Figure 6 and Figure 7). At the final MRI follow-up for each CCM, the same result was observed in the 10th post-radiosurgical year.

### 3.4. Post-Radiosurgical Perifocal Brain Edema

A total of 14 (46.7%) perifocal brain edema events were noted during the follow-up period, namely 3 (10%) symptomatic events and 11 (36.7%) asymptomatic events. Among all the edema events, the median time interval from LINAC SRS to the occurrence of perifocal brain edema was 12 (range: 3–70) months. The median duration of the edematous change was 28.5 (range: 6–141) months. Seven persistent edematous changes were noted until the last MRI follow-up, and all of them were asymptomatic.

All three symptomatic perifocal brain edema events occurred within the first 2 years after LINAC SRS, and neither of them persisted permanently. The median time interval between LINAC SRS and symptomatic perifocal brain edema events was 10 (range: 6–15) months. All three events resulted in a large edematous change on MRI. Two patients received glucocorticoid therapy for 2 months, and one patient received glucocorticoid therapy for 4 months (Figure 8). All three patients returned to their baseline neurologic condition afterward. Three post-radiosurgical cyst formations were noted. Two of them were in remission in the 3rd and 6th years, respectively, whereas the other one remained for 13 years in the last MRI follow-up.

In terms of prognostic factors, the univariate analysis revealed four potentially hazardous variables (*p* < 0.2) for all post-radiosurgical perifocal brain edema events (both symptomatic and asymptomatic events), including a deep CCM location, a longer interval between the first bleeding event and radiosurgery treatment, a lesion volume of >3 cm^3^, and the presence of developmental venous anomaly (DVA). After adjustment for confounding factors in the multivariate analysis, a lesion volume of more than 3 cm^3^ and the presence of DVA were significantly hazardous. By contrast, the univariate analysis for symptomatic perifocal brain edema events revealed no potentially hazardous variables. Therefore, multivariate analysis was not performed. In addition, all patients who developed symptomatic edema events received a marginal dose of >1600 cGy had a deep CCM. There was no underlying hypertension, cerebrovascular disease, or DVA. Thus, these analyses could not be performed. The results are summarized in Figure 9 and Figure 10.

## 4. Discussion

### 4.1. Evidence of SRS in Bleeding Control for CCMs

Unlike in arterio-venous malformation or DVAs, in which post-treatment efficacy can be directly evaluated through angiography, the efficacy of SRS in treating CCMs can only be evaluated on the basis of rebleeding events. Three meta-analyses investigated the AHR of untreated CCMs without previous bleeding events. Gross et al. reported an AHR of 2.5% (95% confidence interval [CI] = 1.3–5.1%) [7]. Horne et al. indicated a 5-year hemorrhage risk of 15.8% (95% CI = 13.7–17.9%) [6]. Taslimi et al. pooled patients into homogenous groups to report more specific hemorrhage rates and discovered the AHRs of asymptomatic CCMS to be 0.3% (95% CI = 0.1–0.5%) and 2.8% (95% CI = 2.5–3.3%) per patient-year for non-brainstem and brain stem CCMs, respectively [8]. The study also observed that the AHR of CCMs with at least 1 bleeding event increased to 6.3% (95% CI = 3–13.2%) and 32.3% (95% CI = 19.8–52.7%) in non-brainstem and brainstem lesions, respectively. 

A previous bleeding event and a brainstem location are two risk factors for future CCM hemorrhage that have been identified in several studies [8,14,15,16] and conclusively by an individual patient data meta-analysis [6]. One prospective study reported that patients with Zabramski class I and II lesions had a significantly higher AHR [17]. Other risk factors remained controversial. 

One meta-analysis reported that the AHR following SRS was 2.40% (95% CI = 2.05–2.80%) [18]. Karaaslan et al. demonstrated a significant decrease in AHR from 15.3% to 2.6% during the first 2 years after treatment and to 1.4% thereafter [19]. The guidelines for treating CCMs with SRS proposed by Niranjan et al. encourage selecting patients on the basis of age, location, history of hemorrhage, and surgical resection risk [20]. In the present study, the overall AHR following LINAC SRS was 4.5%.

To the best of our knowledge, no prior study has determined prognostic factors for post-radiosurgery CCM rebleeding. This study revealed two variables to be significantly protective against rebleeding after SRS treatment, namely a marginal dose of >1600 cGy (*p* = 0.033) and a shorter time interval between diagnosis and SRS (*p* = 0.020). Old age was a protective factor, whereas a large lesion volume was hazardous in the univariate analysis. However, neither of them exhibited significance after adjustment for confounding in the multivariate analysis.

### 4.2. Post-Radiosurgical Response in Volume Change

Changes in the volume and appearance of CCMs are common in imaging studies. The dynamic appearance can be explained by hemorrhage and hemorrhage resolution [21]. Therefore, apart from bleeding control, a stable or reduced lesion size is a goal for treating CCMs. A meta-analysis conducted by Wen et al. revealed that six of nine studies reported a reduction in volume, ranging from 10.2% (4 in 39 patients) to 59.8% (49 in 82 patients) [22]. Another meta-analysis conducted by Kim et al. reported a lesion volume reduction in 47.3% of patients and stationary lesions in 49.4% on the last follow-up images [23]. 

In the present study, we try to evaluate the volume response of CCMs to SRS treatment. Lesion control was defined as complete remission, partial remission, or no change in terms of tumor volume. A total of 23 (88.5%) CCMs exhibited volume control within 2 years after SRS and 25 (96.2%) within 10 years. Two CCMs reached CR. In addition, a prospective study conducted by Clatterbuck et al. revealed no relationship between the lesion size and hemorrhage rate [21]. Although the results of this study and the current evidence support the efficacy of SRS in CCMs volume stabilization and reduction, further studies are warranted to evaluate the relationship between variations in the lesion volume and the risk of post-radiosurgical repeated hemorrhage. 

### 4.3. Post-Radiosurgical Adverse Radiation Effects

The possible mechanism of radiation-induced brain injury with the neuroinflammation cascade has been identified in animal studies [24]. The cascade involves several concomitant processes, including blood–brain barrier (BBB) disruption [25,26], neural progenitor cell death [27,28], neurogenesis inhibition in the hippocampus [29], and direct activation of glia resulting in senescence-associated secretory phenotype cytokines [30,31], making edematous change a frequent presentation after radiation therapy to brain lesions.

In this study, 14 patients developed post-radiosurgical perifocal brain edema on MRI. The overall incidence was 46.7%. However, only three of them were symptomatic and required further medical treatment. The incidence of symptomatic events was 10%. According to the previous studies, SRS-related cerebral edema is associated with tumor size, lesion location, prescribed radiotherapy dose, and edema presence before treatment, but the results have been inconsistent [32,33,34]. The results in the present study indicate that a lesion volume of >3 cm^3^ (*p* = 0.017) and a coexisting DVA (*p* = 0.006) are significantly hazardous for all post-radiosurgery perifocal brain edema. No significant risk factor was identified for symptomatic events.

Larger CCMs have been demonstrated to be hazardous for post-radiosurgery edema in multiple studies, but this finding remains controversial [34,35]. The results proposed by Niranja and Lunsford suggest that the volume of the brain tissue receiving 12 Gy or more provides predictive information for post-radiosurgery imaging changes [20]. The coexistence of CCM and DVA was the most common mixed vascular malformation, with an incidence of 13–40%. The pathophysiology is still uncertain, but they tend to coexist in the same territory. In addition, DVAs might play an evolutionary role in the development and growth of CCMs [36]. DVAs are generally treated conservatively because of their contribution to normal venous drainage and the risk of venous infarction [37]. However, with a thin tunica media, which is believed to contribute to post-radiosurgical vascular sclerosis, smooth muscle hypertrophy, and obliteration [38,39], SRS should exert a weak effect on DVAs. Studies should determine whether a coexisting DVA carries an increasing risk of post-radiosurgical perifocal brain edema. Further investigation is also necessary to elucidate the underlying pathophysiology. 

In terms of the timeline of SRS-related cerebral edema, Harat et al. observed the most prominent edematous effect at 6 months after SRS, with an average duration of 15 months in 34 patients receiving LINAC SRS for various diagnoses [34]. In this study, the median time for post-radiosurgical perifocal brain edema was 11 months. The median duration was 28.5 (range: 6–141) months but should have been longer because of the seven persistent AREs observed on their last MRI follow-ups.

### 4.4. Optimal Marginal Dose for CCMs

The optimal dose of SRS for CCMs remained controversial and differs when applied to different locations. Hasegawa et al. reviewed 12 clinical studies and suggested that low doses (≤1500 cGy) might provide effectiveness similar to that of higher doses but with a lower risk of AREs [14,35,40,41,42,43,44,45,46,47,48,49,50]. However, the effectiveness was compared among studies with different patient characteristics, protocols, and techniques. In addition, most studies have focused on brainstem CCMs, which share patient characteristics with those in our study. In this study, the authors observed that a higher dose, with a cutoff of >1600 cGy, significantly prevented rebleeding in non-brainstem CCMs without causing a significantly higher incidence of symptomatic perifocal brain edema. Currently, the authors treat the non-brainstem CCMs with a marginal dose between 16 and 18 Gy to reach a lower incidence of post-radiosurgical AHR and a better volume control.

### 4.5. Optimal Timing of SRS Treatment

SRS should be performed for sporadic CCMs (1) with multiple bleeding episodes or (2) in a critical location with one bleeding episode. CCMs present in a non-critical location when only one bleeding episode is observed [20]. Another study suggested that even patients with brainstem CCMs who have experienced only a single bleeding event should not be contraindicated for SRS [36]. In our clinical practice, we treated patients with CCMs even after one bleeding event. The results indicate a significantly lower risk of rebleeding with a shorter time interval between the first symptomatic bleeding event of CCMs and SRS treatment. Studies should more strictly select their populations and protocols to further investigate this finding.

### 4.6. LINAC SRS versus GKRS

Compared with GKRS, LINAC SRS is advantageous, with its relative cost-effectiveness and wider range of applications. Although LINAC SRS is sometimes regarded as inferior in terms of dose conformity and accuracy, with advances in software and computer technology, a decreasing incidence of post-radiosurgery neurological complications is expected [51].

As to GKRS, Poorthuis et al. reported a post-radiosurgical annual incidence of 2.40% (95% CI = 2.05–2.80) for symptomatic ICH and 0.71% (95% CI = 0.53–0.96) for non-hemorrhagic persistent focal neurologic deficits [18]. The meta-analysis published by Kim et al. included 559 patients with brainstem CCMs treated with GKRS, and 17 treated with LINAC SRS [23]. They reported lesion volume reducing in 47.3% of the patients and being stationary in 49.4% on the last follow-up images. By contrast, our study reported an overall 4.5% incidence of CCM hemorrhage after LINAC SRS. Of the patients, 46.7% encountered post-radiosurgical perifocal edema, but only 10% were symptomatic. No persistent neurologic deficit was observed during the follow-ups. In addition, 14 out of 26 CCMs (53.8%) showed reduction in volume, and 11 out of 26 CCMs (38.5%) were stationary.

As to the lower post-radiosurgical AHR reported by Poorthuis et al., the authors suggested that the finding could be attributable to the inclusion of patients with asymptomatic CCMs in their study design [18]. Similar results in persistent neurologic deficit and volume control were revealed in this study. Therefore, the authors suggested that LINAC can be a favorable option for SRS for treating symptomatic non-brainstem CCMs in medical institutions without a GKRS system.

### 4.7. Strengths and Limitations

Most studies have examined the efficacy of SRS by comparing pre-radiosurgical and post-radiosurgical outcomes. Although several studies have investigated risk factors for CCM bleeding, no study has investigated prognostic factors for post-radiosurgical rebleeding. In addition, evidence for the use of LINAC SRS for treating CCMs is lacking. This study provides reference for treatment planning.

This study has some limitations. First, this is a single-center retrospective study with a small sample. In addition, significant heterogeneity in the follow-up interval and duration due to the small sample and retrospective design may have contributed to selection and observation bias. To minimize this bias, analyses were adjusted as described.

## 5. Conclusions

This study demonstrated the efficacy of LINAC SRS in preventing CCM rebleeding. An early LINAC SRS intervention with a higher marginal dose (>1600 cGy) might provide better bleeding control for non-brainstem CCMs of <3 cm^3^ without a coexisting DVA. In addition, the risk of perifocal brain edema was not significantly higher. Further investigation is warranted.

## Figures and Tables

**Figure 1 life-12-01363-f001:**
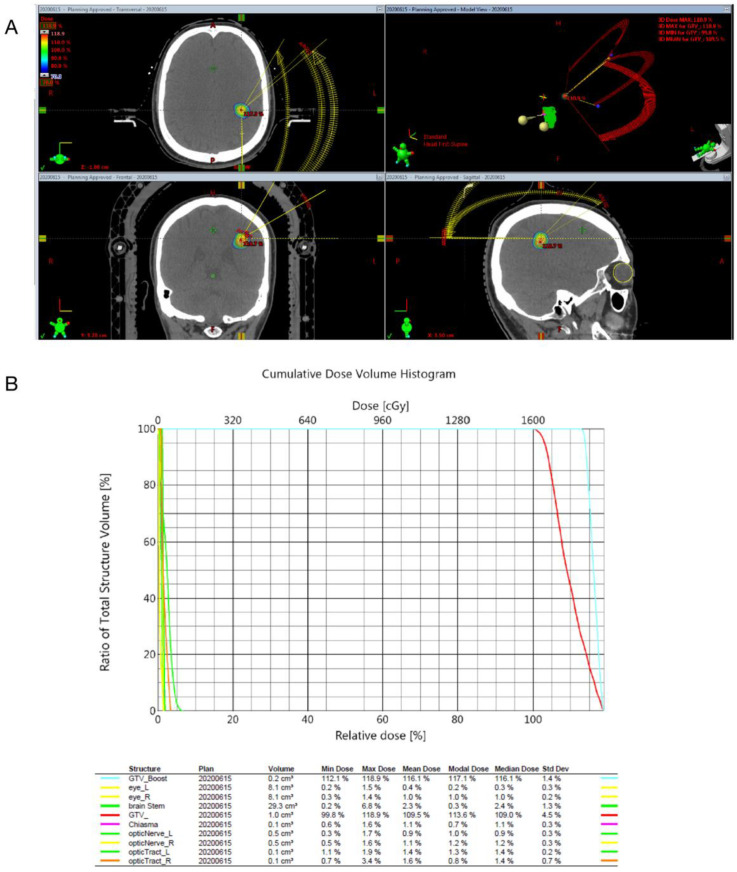
(**A**) Isodose line covering the tumor for radiosurgery performed with the Novalis system (BrainLAB, Heimstetten, Germany). (**B**) Cumulative dose–volume histogram indicating 99.8% of the target volume being covered with 100% of the isodose line.

**Figure 2 life-12-01363-f002:**
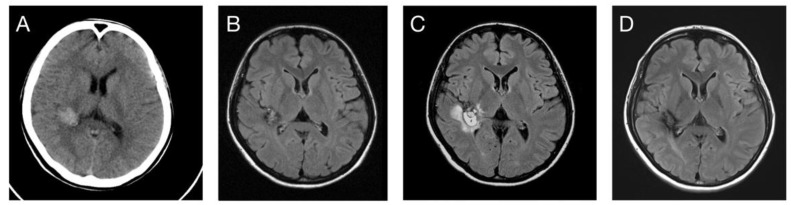
(**A**,**B**) A 22-year-old female patient developed spontaneous intracranial hemorrhage due to right deep temporal cavernoma. (**C**) Rebleeding occurred 34 months after SRS treatment. (**D**) The tumor exhibited a slight decrease in volume after a 130-month follow-up without additional bleeding episodes.

**Figure 3 life-12-01363-f003:**
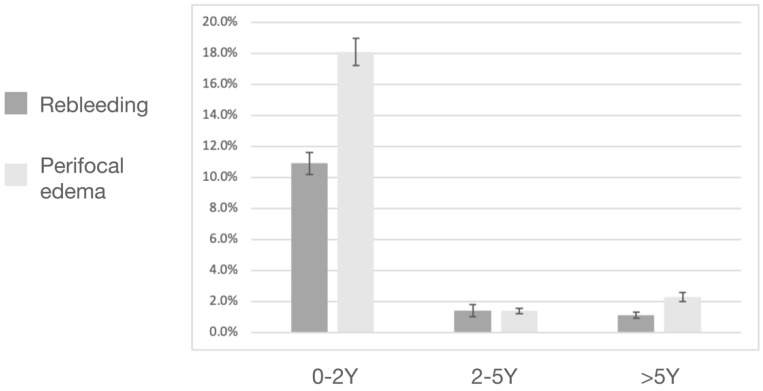
Annual incidence of post-radiosurgical rebleeding and perifocal brain edema events 0–2, 2–5, and 5 years later.

**Figure 4 life-12-01363-f004:**
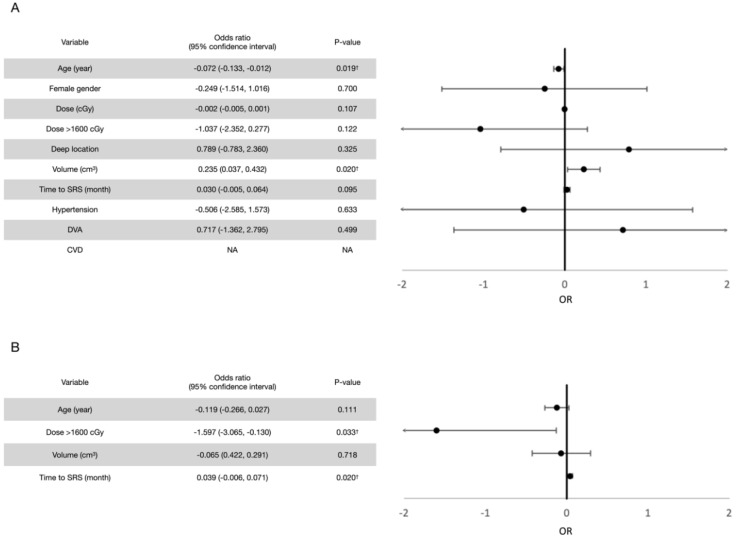
(**A**) Univariate analysis for prognostic factors for all rebleeding events after LINAC SRS treatment. Variables with a *p* value of <0.2 in univariate analyses were introduced to the multivariate analysis. (**B**) Multivariate analysis for prognostic factors for all rebleeding events after LINAC SRS treatment. † Variables with a *p* value of <0.05.

**Figure 5 life-12-01363-f005:**
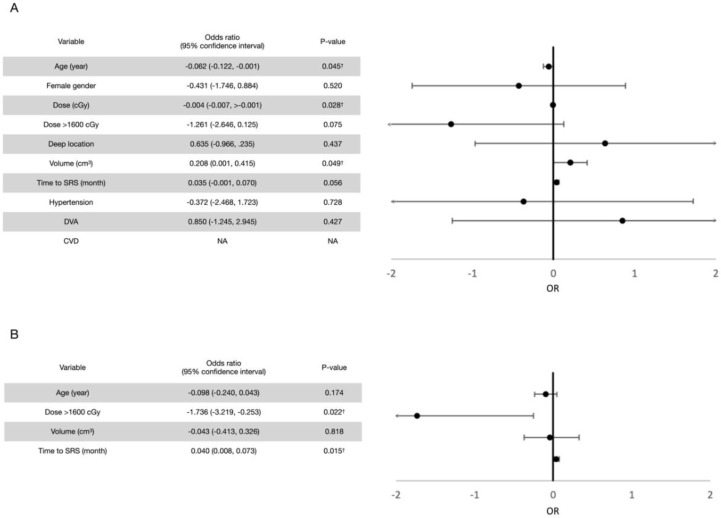
(**A**) Univariate analysis for prognostic factors for symptomatic rebleeding events after LINAC SRS treatment. Variables with a *p* value of <0.2 in univariate analyses were introduced into the multivariate analysis. (**B**) Multivariate analysis for prognostic factors for symptomatic rebleeding events after LINAC SRS. † Variables with a *p* value of <0.05.

**Figure 6 life-12-01363-f006:**
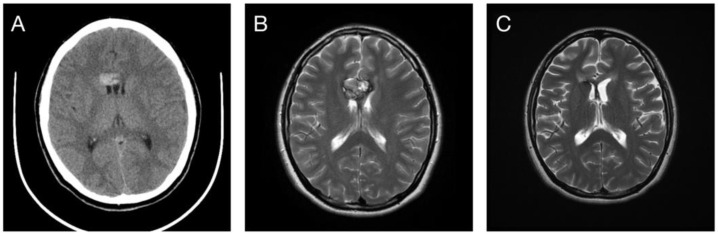
(**A**) A 21-year-old female patient suddenly developed a severe headache and nausea. CT revealed an intracranial hematoma in the anterior corpus callosum. (**B**) MRI 3 months after CT revealed a cavernoma, and SRS treatment with 2000 cGy was applied. (**C**) The tumor exhibited remission completely 24 months after SRS treatment.

**Figure 7 life-12-01363-f007:**
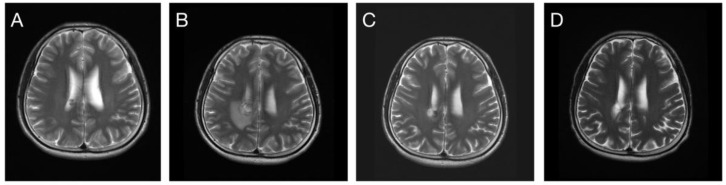
(**A**) A 36-year-old male patient experienced a severe headache accompanied by nausea and vomiting. A cavernoma with bleeding located in the right posterior paraventricular area was diagnosed after CT and MRI. SRS treatment with 16 Gy was administered. (**B**) Repeated bleeding accompanied with perifocal brain edema occurred 9 months after SRS. (**C**) No medication was administered because of the absence of clinical symptoms, and edema subsided 12 months after SRS. (**D**) Complete tumor remission was noted at the MRI follow-up 60 months later.

**Figure 8 life-12-01363-f008:**
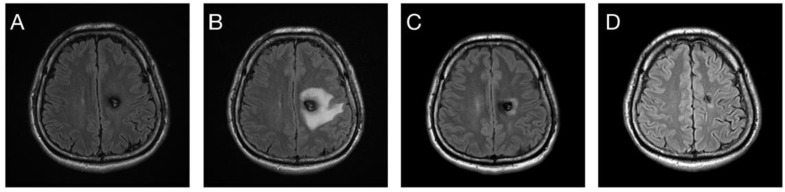
(**A**) A 41-year-old male patient who had a left posterior frontal cavernoma received SRS treatment with 20 Gy. (**B**) Severe perifocal brain edema attack 10 months after SRS induced right hemiparesis clinically. (**C**) Steroids were administered for 4 months to control edema, which subsided gradually in 15 months. (**D**) Decrease in tumor size was noted after 41-month follow up.

**Figure 9 life-12-01363-f009:**
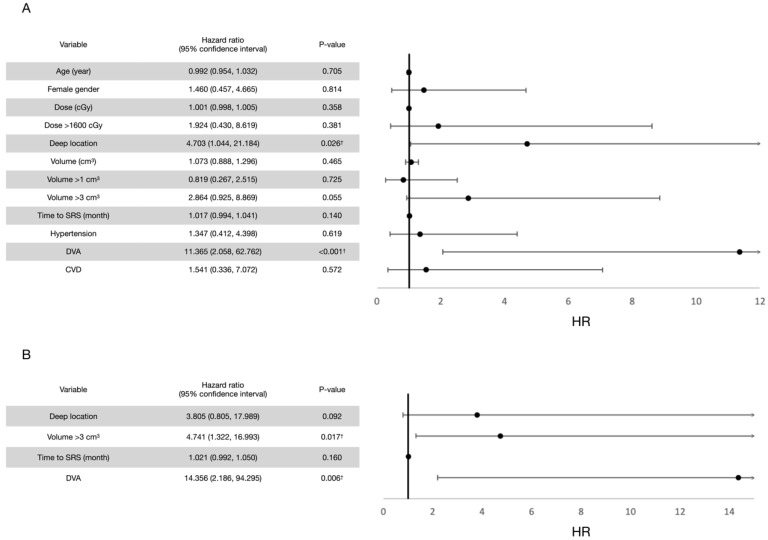
(**A**) Univariate analysis for prognostic factors for all perifocal brain edema events after LINAC SRS treatment. Variables with a *p* value of <0.2 in univariate analyses were introduced to the multivariate analysis. (**B**) Multivariate analysis for prognostic factors for all perifocal brain edema events after LINAC SRS treatment. † Variables with a *p* value of <0.05.

**Figure 10 life-12-01363-f010:**
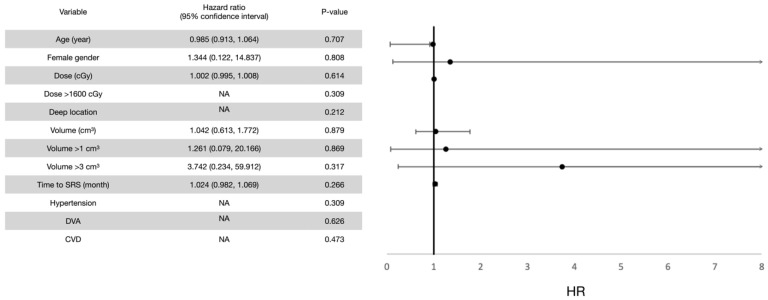
Univariate analysis for prognostic factors for symptomatic perifocal brain edema events after LINAC SRS treatment. The multivariate analysis was not performed because no variable exhibited potential significance (*p* < 0.2) in the univariate analysis.

**Table 1 life-12-01363-t001:** Characteristics of 30 patients receiving linear accelerator stereotactic radiosurgery for sporadic, solitary cerebral cavernous malformations with symptomatic bleeding events.

	Number (%) or Median (Standard Deviation)
Gender	
Male	12 (40%)
Female	18 (60%)
Age (year)	43 (15.2)
Initial Zabramski classification	
Class I	4 (13.3%)
Class II	25 (83.3%)
Class III	1 (3.3%)
Class IV	0 (0%)
Location	
Non-eloquent WGJ	9 (30%)
Eloquent WGJ	1 (3.3%)
Subcortical nucleus	4 (13.3%)
White matter	15 (50%)
Cerebellum	1 (3.3%)
Initial symptoms	
Motor deficit	11 (36.7%)
Headache	8 (26.7%)
Seizure	8 (26.7%)
Dizziness	6 (20%)
Cranial nerve deficit	5 (16.7%)
Sensory deficit	1 (3.3%)
Time to treatment (months)	11 (20.8)
Marginal dose (cGy)	2000 (201.9)
Initial volume (cm^3^)	0.97 (5.67)
Follow-up period (months)	
Total	61 (63.0)

Abbreviations and explanations: WGJ, white-gray matter junction; cGy, centigray; time to treatment, the interval between first symptomatic bleeding event and radiosurgery treatment; MRI, magnetic resonance imaging.

## Data Availability

All relevant data are available on request.

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
