# Peer review of "Prognostic Factors Analysis for Intracranial Cavernous Malformations Treated with Linear Accelerator Stereotactic Radiosurgery"

_life, 2022, doi:10.3390/life12091363_

Round 1

Reviewer 1 Report

The introduction is appropriate, the methodology is exhaustive, and the results are well-presented. The discussion is adequate.

I think the font size of the text in figures 1, 4, 5, 9 and 10 are too small to read.

The references are current.

Overall, the manuscript is well-written.

Author Response

Thanks for your review and advice. Our reply was presented in the word file. 

Reviewer 2 Report

The authors analyzed 30 patients with cranial cavernous malformations treated with linear accelerator stereotactic radio surgery in a single center. 

The analysis of risk factors and outcome predictors based on clinical and imaging data is straight forward and provides information for clinical practice that can be validated in a larger study. 

There are some improvements that should be made before publication, in detail:

-       Line 33: add “primary objective of treatment”

-       Figure 1: please enlarge the histogram

-       Please give a short overview over the used MRI protocol and the system (1.5 T?)

-       Line 105: please use consistent terms, tumor could be misleading here

-       Were patients treated with LINAC SRS after interdisciplinary board discussion? Please add a short explanation on the algorithm used in your center 

-       Could you please comment on the long average time between bleeding event and treatment?

-       Figure 2: please use the T2 image for D as well 

-       Figure 4 & 5 & 9 & 10 are hardly readable, please increase quality and size of the tables

-       Figure 8: use FLAIR image for B as well

-       Please add an explanation for dose finding for the individual patient at your clinic, did you now switch to >1600cGy in clinical routine?

-       Paragraph 4.6: you have not compared this in the presented study, this needs to be clearer differentiated and discussed in more detail

Author Response

Thanks for your review and advice. Please see the attachment for our reply.
